# Denouement of the Energy-Amplitude and Size-Amplitude Enigma for Acoustic-Emission Investigations of Materials

**DOI:** 10.3390/ma15134556

**Published:** 2022-06-28

**Authors:** Sarah M. Kamel, Nora M. Samy, László Z. Tóth, Lajos Daróczi, Dezső L. Beke

**Affiliations:** 1Department of Solid State Physics, University of Debrecen, P.O. Box 400 Debrecen, Hungary; sarahkamel@scicence.unideb.hu (S.M.K.); noramohareb88@gmail.com (N.M.S.); toth.laszlo@scicence.unideb.hu (L.Z.T.); lajos.daroczi@scicence.unideb.hu (L.D.); 2Physics Department, Faculty of Science, Ain Shams University, Cairo 11566, Egypt; 3Department of Physics, Faculty of Education, Ain Shams University, Cairo 11341, Egypt

**Keywords:** shape memory alloys, acoustic emission, scaling relations, temporal shapes of avalanches

## Abstract

There are many systems producing crackling noise (avalanches) in materials. Temporal shapes of avalanches, *U*(*t*) (*U* is the detected voltage signal, *t* is the time), have self-similar behaviour and the normalized *U*(*t*) function (e.g., dividing both the values of *U* and *t* by *S*^1/2^, where *S* is the avalanche area), averaged for fixed S, should be the same, independently of the type of materials or avalanche mechanisms. However, there are experimental evidences that the temporal shapes of avalanches do not scale completely in a universal way. The self-similarity also leads to universal power-law-scaling relations, e.g., between the energy, *E*, and the peak amplitude, *A_m_*, or between *S* and *A_m_*. There are well-known enigmas, where the above exponents in acoustic emission measurements are rather close to 2 and 1, respectively, instead of $E\sim A_{m}^{3}$ and $S\sim A_{m}^{2}$, obtained from the mean field theory, MFT. We show, using a theoretically predicted averaged function for the fixed avalanche area, $U\left(t\right)=at\exp\left(-bt^{2}\right)$ (where *a* and *b* are non-universal, material-dependent constants), that the scaling exponents can be different from the MFT values. Normalizing *U* by *A_m_* and *t* by *t_m_* (the time belonging to the *A_m_*: rise time), we obtain $t_{m}\sim A_{m}^{1-\varphi }$ (the MFT values can be obtained only if *φ* would be zero). Here, *φ* is expected to be material-independent and to be the same for the same mechanism. Using experimental results on martensitic transformations in two different shape-memory single-crystals, *φ* = 0.8 ± 0.1 was obtained (*φ* is the same for both alloys). Thus, dividing *U* by *A_m_* as well as *t* by $A_{m}^{1-\varphi }$ (~*t_m_*) leads to the same common, normalized temporal shape for different, fixed values of *S*. This normalization can also be used in general for other experimental results (not only for acoustic emission), which provide information about jerky noises in materials.

## 1. Introduction

It is well-known that many systems produce crackling noise (avalanches) with power-law-distribution characteristics (i.e., the probability-distribution-density function (PDF), *P*(*x*), can be given as $P\left(x\right)\sim x^{-\eta }\exp\left(-\frac{x}{x_{c}}\right)$, where *x* can be the peak amplitude, *A_m_*, size, *S*, energy, *E*, or duration, *T*; *η* is the characteristic exponent, and *x_c_* is the cut-off value) [1,2,3,4,5,6,7,8,9]. The power-law distributions reflect a self-similar behaviour spanning wide range of the parameter, *x* (e.g., the temporal shape of an avalanche looks the same at different time scales). Examples for such behaviour can be the classical Barkhausen noise, sand piles, fracture, martensitic transformations in shape memory materials, plastic deformations, etc. In many cases, the avalanches are jerky responses to slowly changing driving force or field. Thus, considerable efforts were devoted to predict how the corresponding exponents of the above distributions can be grouped into universality classes [5,10,11]. In addition, power-law-scaling relations between the exponents of the above parameters were obtained (e.g., the energy, *E*, is related to the amplitude, $A_{m}$, as $E\sim A_{m}^{\chi }$) with predictions that these should be the same within one universality class [1,6,12,13,14]. Furthermore, the self-similarity leads not only to power laws, but to universal-scaling functions, which can have predictive power, and in recent publications the authors have gone beyond the power laws and focused on the universal, (properly normalized) temporal shape of avalanches [1,2,4,5,10]. 

For experimental investigations of the above power-law relations or temporal shapes of avalanches, different experimental techniques can be used in which the measured parameter (e.g., the voltage, *U*(*t*), in acoustic emission, AE, or magnetic measurements) is proportional to the corresponding interface velocity *v*(*t*), characteristic for the crackling-noise emission. Until now, Barkhausen noise measurements provided excellent agreement between the predicted [7] and experimentally determined, normalized temporal-avalanche shapes at fixed duration or shape [15]. Besides Barkhausen noise measurements, other techniques such as AE [8,16,17,18,19,20,21,22] or high-resolution detection of the deformation or stress drops during plastic deformation (see e.g., [9,23,24]) are also used, but the agreement with theoretical predictions is far less satisfactory than those obtained from Barkhausen noise investigations. For instance, there are observations that the normalized shapes of avalanches do not collapse on the same reduced curve for different size or duration bins (see e.g., [7,8,9]). There are two factors, which can have an influence on the experimentally determined characteristic parameters of avalanches. The finite value of the threshold, *C*, at small signals can lead to deviations from the predicted behaviour. The effects caused by the transfer properties of the detection system can also cause distortions. The AE signals depends not only on the properties of the source function (e.g., on *v*(*t*)) but also on the macroscopic vibration (ringing) of the sample. This means that the detected signal is the convolution of the source function, *v*(*t*), and the transfer function, which can be taken as a form of damped oscillation [8,17,25]:(1)$$f\left(t\right)=\cos\left(wt\right)\exp\left(-\frac{t}{\tau_{a}}\right),$$
where *τ_a_* is the characteristic attenuation time of the signal, and *ω* is the resonant frequency. Thus, it was concluded in [25] that the measured AE spectrum does not reflect the temporal shape of avalanches (i.e., the *v*(*t*) distribution) nor the model predictions. Therefore, a detailed analysis of the observed AE jerk profiles only reveals information about the transfer function of the measuring system (material properties + detector: see Figure 1 in [25]) and says little about the local avalanche mechanism. It was also shown, from the convolution of the transfer function with different model functions for the source [25], that while the characteristic exponents of the energy and size PDF’s were invariant, the detected duration time, D, was significantly distorted compared to the true duration time, *T*. Furthermore, the so-called energy-size enigma was exposed: while the mean field theory (MFT) predicts *χ* = 3 for the scaling exponent between the energy and amplitude, their model simulations for AE results provided *χ* = 2. It is worth mentioning that in a set of papers by Barcelona’s group [17,18,19] a less pessimistic conclusion was drawn. It was argued that if one considers the convolution of a simple rectangular-signal source, then for signals with long duration times, T, as compared to $\tau_{a}$, i.e., $\frac{\tau_{a}}{T}\ll 1$, the detected duration time can be close to the true duration time, T. On the other hand, for $\frac{\tau_{a}}{T}\gg 1$, the results provide information about the attenuation time, $\tau_{a}$ [17,18,19], and the scaling exponents between the energy and duration time or the energy and amplitude are considerably different from the values predicted by the MFT. For intermediate values of $\frac{\tau_{a}}{T}$ a transition between the above two limits can be observed. On the other hand, surprisingly, as it was also mentioned in [25], it was obtained that even for $\frac{\tau_{a}}{T}\ll 1$, $E\sim A_{m}^{2}$ was observed. 

In this communication we will start from the theoretically predicted form of the avalanche’s temporal shape [7,8,9] for a fixed avalanche area, $U\left(t\right)=at\exp\left(-bt^{2}\right)$ (where *a* and *b* are non-universal, material-dependent constants). We will show that, e.g., deviations from the universal, normalized avalanche shape can be described by a parameter, *φ*. *φ* is the exponent describing the relation between the two recently proposed scaling parameters [25], the maximum voltage, *A_m_*, and the maximum time (raising time), *t_m_*. We assume that their ratio, $\frac{A_{m}}{t_{m}}$, instead of being constant, is given as $\frac{A_{m}}{t_{m}}\sim A_{m}^{\varphi }$. *φ* is material-independent, and the same is the same for the same mechanism. It appears, as a correction term, in the scaling exponents, and, thus, provides the denouement of the enigmas. Thus, we will illustrate, using experimental data obtained during martensitic transformation in two ferromagnetic shape-memory alloys, that this indeed leads to deviations from the predicted scaling exponents and, e.g., the slopes of the *logS* versus *logA_m_* or *logE* versus *log**A_m_* are given by 3−φ and 2−φ, respectively, i.e., they can be much smaller than the predicted MFT values (2 and 3). Furthermore, dividing *U*(*t*) by *A_m_* as well as *t* by $A_{m}^{1-\varphi }$ (~*t_m_*) leads to the same common, normalized temporal shape for different fixed values of *S*. Our results can also be valid in general for scaling relations between the experimentally determined parameters from other types of measurements of avalanches (magnetic emission [26], high-resolution detection of the deformation or stress drops, etc.) and not only for AE. 

We will also demonstrate that, by using a properly chosen scaling window and taking into account corrections determined by the parameters, *φ*, $\frac{\tau_{a}}{T}$, and $\frac{C}{A_{m}}$, the scaling relations are in excellent agreement with the predictions of the MFT. The lower and upper bounds of such a scaling window are related to the combined conditions of the $\frac{C}{A_{m}}\ll 1$ and $\frac{\tau_{a}}{T}\ll 1$, as well as to the possible overlaps of avalanches and/or to small numbers of hits, respectively. 

## 2. Expressions for the Exponents of Scaling Relations

Self-similarity (see e.g., [1,2]) implies that the average temporal shape *v*(*t*) ~ *U*(*t*) of avalanches scales in a universal way. For instance, in [2] the average of *U*(*t*) (averaged over avalanches of fixed durations, *T*) was considered, if the time scale was reduced by *T*:(2a)$$\langle U\left(T,t\right)\rangle =u\left(T,\frac{t}{T}\right),$$
and $u\left(T,\frac{t}{T}\right)$ was compared to itself on a slightly increased time scale. It was obtained, that
(2b)$$u\left(T,\frac{t}{T}\right)=u_{o}\left(\frac{t}{T}\right)T^{b},$$
i.e., if the voltage is scaled by *T^b^*, then the scaling function, $u_{o}\left(\frac{t}{T}\right)$, is a universal theoretical prediction (for large sizes and long times) [1,2]:(3)$$\langle U\left(T,\frac{t}{T}\right)\rangle =u_{o}\left(\frac{t}{T}\right)T^{b}.$$

From the definition of the avalanche size, *S*, we obtain:(4)$$S=\int_{0}^{T}T^{b}u_{o}\left(\frac{t}{T}\right)dt=T^{1+b}\int_{0}^{1}u_{o}\left(\frac{t}{T}\right)d\left(\frac{t}{T}\right)=T^{1+b}F_{S},$$
i.e., since the integral is a constant (*F_S_* = *const.*) for universal $u_{o}\left(\frac{t}{T}\right)$, we arrive at the scaling relation
(5)$$S\sim T^{b+1}=T^{\gamma }.$$

In theoretical papers, derivations for the power exponent, *γ*, are given and, e.g., the mean field approximation [6] gives that *γ* = 2.

In order to obtain an exponential relation between the average voltage, *U_av_*, and *T*, we can write from (4)
(6)$$U_{av}=\frac{S}{T}\sim T^{\gamma -1}.$$

Similarly, as for the avalanche size, using the definition of the energy of the detected signal, *E*, (i.e., $E\sim \int_{0}^{T}U{\left(t\right)}^{2}dt$), we have
(7)$$E\sim \int_{0}^{T}T^{2\left(\gamma -1\right)}u_{o}{}^{2}\left(\frac{t}{T}\right)dt=T^{2\gamma -1}\int_{0}^{1}u_{o}{}^{2}\left(\frac{t}{T}\right)d\left(\frac{t}{T}\right)=T^{2\gamma -1}F_{E}\sim T^{2\gamma -1}.$$

Note that (5)–(7) are the usual exponent relations between the corresponding quantities [1,2,3,6,10,16,23]. 

Now, from (5)–(7), we have
(8)$$S\sim U_{av}{}^{\frac{\gamma }{\gamma -1}},$$
(9)$$E\sim U_{av}^{\frac{2\gamma -1}{\gamma -1}}.$$

Interestingly, from (8) and (9) we obtain
(10)$$\frac{E}{S}\sim U_{av}$$
and the exponent 1 of *U_av_* is independent of *γ*.

Since in experiments, instead of *U_av_*, the maximal value of the voltage (the peak value), *U_m_*, is commonly determined, so the relation between them should be considered. It was argued in [6] that the peak amplitude is a good measure of *U_av_*, but the relation between these two parameters was not checked experimentally. We will show in the next section that they are indeed interrelated, but instead a linear relation
(11)$$U_{av}\sim U_{m}^{z}$$
can hold, with z<1, for finite thresholds. For a demonstration of this, we use the same averaged source function at a fixed area in MF approximation, which was also investigated in [25] (see also, e.g., [7,8,9]):(12)$$U\left(t\right)=ate^{-{\left(\frac{t}{\tau }\right)}^{2}},$$
where *a* and *τ* are non-universal (material-dependent) constants. *τ* is the characteristic time of the avalanche decay [8] (in [25], $b=\frac{1}{\tau^{2}}$ was used instead of *τ*). The maximum of (12) is at
(13)$$t_{m}=\frac{\tau }{\surd 2}=\frac{1}{\sqrt{2b}}$$
and
(14)$$U_{m}=at_{m}e^{-\frac{1}{2}}=Bt_{m},$$
i.e., *U_m_* and *t_m_* are linearly related to each other if *a* (*B*) is constant. This is in accordance with the result of [25], where this relation was analysed by simulations for the fixed value of a (~*B*), and $U_{m}\sim t_{m}^{\xi }$, with *ξ* = 0.95, was obtained. We will show below that the value of *B* has a definite dependence on *U_m_*, $a\sim B\sim U_{m}^{\varphi }$. Dividing both sides of (12) by *U_m_*, and using (14), we obtain the dimensionless (reduced) form of *U* with the two scaling parameters (recommended also in [25], since they are not distorted by transfer effects) *U_m_* and *t_m_* ($U^{*}=\frac{U}{U_{m}}$ and $t^{*}=\frac{t}{t_{m}}$, respectively) as
(15)$$U^{*}\left(t^{*}\right)=e^{\frac{1}{2}}t^{*}e^{-{\left(\frac{t^{*}}{\tau^{*}}\right)}^{2}}=1.65t^{*}e^{-{\left(\frac{t^{*}}{\tau^{*}}\right)}^{2}},$$
where $\frac{t^{*}}{\tau^{*}}=\frac{t}{\tau }$ and $\tau^{*}=\frac{\tau }{t_{m}}=\surd 2$.

The reduced area of the avalanche is given by
(16)$$S^{*}=\frac{S}{U_{m}t_{m}}=\int_{o}^{T^{*}}U^{*}\left(t^{*}\right)dt^{*}=1.65\frac{\tau^{*2}}{2}\left(1-e^{-{\left(\frac{T}{\tau }\right)}^{2}}\right)=1.65\left(1-e^{-\frac{T^{*2}}{2}}\right),$$
from which we obtain the average *U* as
(17)$$U_{av}=\frac{S}{T}=\frac{U_{m}t_{m}}{T}1.65\left(1-e^{-{\left(\frac{T}{\tau }\right)}^{2}}\right)=\frac{U_{m}}{T^{*}}1.65\left(1-e^{-{\left(\frac{T}{\tau }\right)}^{2}}\right)$$

Thus,
(18)$$\frac{U_{av}}{U_{m}}=\frac{1.65}{T^{*}}\left(1-e^{-{\left(\frac{T}{\tau }\right)}^{2}}\right)=\frac{1.65}{T^{*}}\left(1-e^{-\frac{T^{*2}}{2}}\right).$$

This indicates that *U_av_* and *U_m_* are proportional to each other (as predicted by [6]), only if $T^{*}$ is constant. Now, we can calculate the reduced duration time as the difference of the start and finish times ($t_{s}^{*}$ and $t_{f}^{*}$, respectively) given by a fixed threshold value, *C*, from (15)
(19)$$ln\frac{C}{U_{m}}=ln1.65t^{*}-\frac{t^{*2}}{2}.$$

It is clear that there are two solutions, belonging to the start and finish times:(20)$$t_{s}^{*}≅\frac{C}{1.65U_{m}}=\frac{C}{1.65Bt_{m}}$$
as well as
(21)$$t_{f}^{*}≅\sqrt{-2ln\frac{C}{U_{m}}}=\sqrt{2ln\frac{U_{m}}{C}}=\sqrt{2ln\frac{Bt_{m}}{C},}$$
where (14) was also used. Thus, we have
(22a)$$T^{*}≅t_{f}^{*}-t_{s}^{*}=\sqrt{2ln\frac{U_{m}}{C}}-\frac{C}{1.65U_{m}}≅\sqrt{2ln\frac{U_{m}}{C}}=\sqrt{2ln\frac{Bt_{m}}{C},}$$

For $\frac{C}{U_{m}}=10^{-4};\;10^{-3};\;10^{-2};\;10^{-1}$, the first term is 4.3; 3,7; 3.0; and 2.1, respectively, i.e., neglecting the $\frac{C}{1.65U_{m}}$ is reasonable. It is worth noting that the duration time or its reduced value goes to an asymptotic limit as C goes to zero. We use the following notations: *ln* corresponds to *log_e_*, and *log* corresponds to *log*_10_, respectively. Thus, (22a) can also be written as
(22b)$$T^{*}=\sqrt{\frac{2}{log_{10}e}log_{10}\frac{U_{m}}{C}}=\sqrt{\frac{2}{0.434}log\frac{U_{m}}{C}}=\sqrt{4.6log\frac{U_{m}}{C}},$$
where *e* = 2.718, and *log*_10_2.718 = 0.434.

Since *a* (and *τ* or *b*) in (12) are non-universal constants (expressing also that the normalized shapes of avalanches do not fully collapse on the same reduced curve for different size or duration bins), their dependence on the scaling parameters cannot be excluded. We describe this by allowing that the $B=\frac{U_{m}}{t_{m}}$ parameter can be dependent on *U_m_* as: (23)$$B=\alpha U_{m}^{\varphi },$$
where *α* is a proportionality constant. This means that, instead of a linear relation between *U_m_* and *t_m_*, we have
(24)$$U_{m}^{1-\varphi }=\alpha t_{m},$$
which leads to
(25)$$\frac{\partial lnU_{m}}{\partial lnt_{m}}=\frac{1}{1-\varphi }.$$
(see also the next chapter for experimental determination of the value of *φ*).

We can also derive an expression for the exponent of the scaling relation between *U_m_* and *T*, using (22a) and (24) in the form $T=t_{m}T^{*}=\frac{U_{m}}{B}T^{*}=\frac{U_{m}^{1-\varphi }}{\alpha }\sqrt{2ln\frac{U_{m}}{C}}$ as;
(26)$$\frac{\partial lnU_{m}}{\partial lnT}=\frac{\partial lnU_{m}}{\partial U_{m}}\frac{\partial U_{m}}{\partial T}\frac{\partial T}{\partial lnT}=\frac{T}{U_{m}}{\left(\frac{\partial T}{\partial U_{m}}\right)}^{-1}=\frac{1}{1-\varphi +\frac{1}{2ln\frac{U_{m}}{C}}}.$$

Using the above expressions, the slope of the $lnU_{av}$ versus $lnU_{m}$ can also be estimated from (18) as
(27)$$lnU_{av}\sim lnU_{m}-lnT^{*}+ln\left(1-e^{-\frac{T^{*2}}{2}}\right)=lnU_{m}-lnT^{*}+ln\left(1-\frac{C}{U_{m}}\right)\;≅lnU_{m}-lnT^{*}$$
and, thus,
(28)$$\frac{\partial lnU_{av}}{\partial lnU_{m}}≅1+\frac{\partial ln\left(1-\frac{C}{U_{m}}\right)}{\partial lnU_{m}}-\frac{\partial lnT^{*}}{\partial lnU_{m}}=1+\frac{1}{\frac{U_{m}}{C}-1}-\frac{1}{2ln\frac{U_{m}}{C}}≅1-\frac{1}{2ln\frac{U_{m}}{C}}=z^{\prime },$$
where (22b) was also used. Neglecting the $\frac{1}{\frac{U_{m}}{C}-1}$ term means a correction of less than 5% if $\frac{U_{m}}{C}>20$. It can be seen that the slope, i.e., the value of $z^{\prime }$, is close to unity only for very large values of $\frac{U_{m}}{C}$, and it is always less than 1: e.g., for $\frac{U_{m}}{C}=100$ or 10, $z^{\prime }$ = 0.9 or $z^{\prime }$ = 0.89, respectively. Note, the above result is independent of the fact that whether *B*(~*a*) is constant or depends on *U_m_*. 

Furthermore, regarding the detected values of the AE parameters (denoted below by *E_AE_*, *S_AE_*, *A_m_*, *A_av_*, and *D*, instead of $E,\;S,\;U_{m},\;U_{av}$, and *T*), we have to take into account that the transfer effects can distort the values of *A_av_* and *D*. Thus, taking that $A_{av}=\frac{S_{AE}}{D}$, instead of (28), we have (using that $A_{av}=\frac{S_{AE}}{D}=\frac{S}{T}\frac{T}{D}=U_{av}\frac{T}{D}$ and that *U_m_* ~ *A_m_*, i.e., *U_m_* = *δA_m_* with δ≅1 for $\frac{\tau_{a}}{T}\ll 1$ [19])
(29a)$$\frac{\partial lnA_{av}}{\partial lnA_{m}}=\frac{\partial lnU_{av}}{\partial lnU_{m}}+\frac{\partial ln\frac{T}{D}}{\partial lnA_{m}}=1-\frac{1}{2ln\frac{A_{m}}{C}}+\frac{\partial ln\frac{T}{D}}{\partial lnA_{m}}=z$$

Furthermore,
(29b)$$\frac{\partial lnA_{av}}{\partial lnT}=\frac{\partial lnA_{av}}{\partial lnA_{m}}\frac{\partial lnA_{m}}{\partial lnT}=\frac{z}{1-\varphi +\frac{1}{2ln\frac{A_{m}}{C}}}.$$

Since the third term in (29a), in accordance with [18,19], is a function of $\frac{\tau_{a}}{T}$ (T≅D if $\frac{\tau_{a}}{T}ln\frac{A_{m}}{C}\ll 1$) too, and *T* can be expressed as a function of $A_{m}$ (see Equation (6)), we can take that this term is the function of *A_m_* only in a given experiment, where $\tau_{a}$ and *C* are constant. Thus, we can write that
(30)$$\frac{\partial ln\frac{T}{D}}{\partial lnA_{m}}=\frac{\partial lnT}{\partial lnA_{m}}-\frac{\partial lnD}{\partial lnA_{m}}=1-\varphi +\frac{1}{2ln\frac{A_{m}}{C}}-\theta ,$$
and, thus,
(31)z=2−φ−θ,
where (29a) and (30) were used. It can be seen that the reciprocal value of the slope of the $lnA_{m}$ versus lnD experimental plot ($\theta =\frac{\partial lnD}{\partial lnA_{m}}$) can be used as a parameter characterizing the transfer effects; for θ=1, these are neglected. 

Furthermore, from (16), using again that $\left(1-e^{-\frac{T^{*2}}{2}}\right)=1-\frac{C}{U_{m}}$
(32)$$\frac{\partial lnS_{AE}}{\partial lnA_{m}}=\frac{\partial lnS}{\partial lnU_{m}}=1+{\left(\frac{\partial lnU_{m}}{\partial lnt_{m}}\right)}^{-1}+\frac{\partial ln\left(1-\frac{C}{U_{m}}\right)}{\partial lnU_{m}}=2-\varphi +\frac{1}{\frac{A_{m}}{C}-1}≅2-\varphi ,$$
or
(33)$$\frac{\partial lnS_{AE}}{\partial lnA_{av}}=\frac{\partial lnS}{\partial lnU_{av}}=\frac{\partial lnS}{\partial lnU_{m}}\frac{\partial lnA_{m}}{\partial lnA_{av}}≅\frac{2-\varphi }{z},$$
and
(34)$$\frac{\partial lnS_{AE}}{\partial lnD}=\frac{\partial lnS}{\partial lnA_{m}}\frac{\partial lnA_{m}}{\partial lnD}=\frac{2-\varphi }{\theta }.$$

For scaling relations containing the energy, we can start from (7) and (15), i.e.,
(35a)$$E\sim U_{m}^{2}t_{m}\int_{o}^{T^{*}}t^{*2}exp\left(-t^{*2}\right)dt^{*}=U_{m}^{2}t_{m}I.$$

The *I* integral has the form:$$\int_{0}^{T^{*}}t^{*2}exp\left(-t^{*2}\right)dt^{*}={\left[-\frac{1}{2}t^{*}exp\left(-t^{*2}\right)+\frac{\sqrt{\pi }}{4}erft^{*}\right]}_{0}^{T^{*}},$$
i.e.,
(35b)$$I=-\frac{1}{2}T^{*}\exp\left[-{\left(T^{*}\right)}^{2}\right]+\frac{\sqrt{\pi }}{4}erf\left(T^{*}\right)=-\frac{1}{2}\left(\sqrt{2ln\frac{U_{m}}{C}}\right){\left(\frac{C}{U_{m}}\right)}^{2}+\frac{\sqrt{\pi }}{4}erf\sqrt{2ln\frac{U_{m}}{C}}.$$

It can be seen that the value of *I* is always positive (*T* > *t_m_*, i.e., $T^{*}>0$). Indeed, for *T** > 2 we can take into account that *erf*2 ≅ 1, so the first term can be neglected as compared to $\frac{\sqrt{\pi }}{4}$ (*T** *exp*(−*T**^2^) = 0.037 for *T** = 2). Thus, $T^{*}>2$ also means that in $\frac{\sqrt{\pi }}{4}\left(1-\frac{2}{\sqrt{\pi }}T^{*}exp\left(-T^{*2}\right)\right)$ the second term is less than 0.05, and we obtain $I≅const=\frac{\sqrt{\pi }}{4}$. The $T^{*}>2$ requirement leads also to the condition that $\frac{U_{M}}{C}>8$ (see also (22)). Thus, we have
(36)$$E\sim U_{m}{}^{2}t_{m}I=\frac{U_{m}^{3}I}{B}=B^{2}t_{m}^{3}I≅B^{2}T^{3}I{\left(2ln\frac{U_{m}}{C}\right)}^{-\frac{3}{2}},$$
and, thus,
(37)$$\frac{\partial lnE}{\partial lnU_{m}}=\frac{\partial lnE_{AE}}{\partial lnA_{m}}\sim 3-\varphi$$
as well as
(38)$$\frac{\partial lnE_{AE}}{\partial lnA_{av}}=\frac{\partial lnE}{\partial lnU_{av}}=\frac{\partial lnE}{\partial lnU_{m}}\frac{\partial lnU_{m}}{\partial lnU_{av}}\sim \left(3-\varphi \right)\frac{1}{z}.$$

Furthermore, the relations between the $\frac{E}{S}$, $\frac{E}{D}$, and $\frac{S}{D}$ ratios and the amplitude can be given as follows.
(39)$$\frac{\partial ln\frac{E_{AE}}{S_{AE}}}{\partial lnA_{m}}=\frac{\partial ln\frac{E}{S}}{\partial lnU_{m}}\sim 1-\frac{\partial ln\left(1-\frac{C}{U_{m}}\right)}{\partial lnU_{m}}=1-\frac{1}{\frac{A_{m}}{C}-1}≅1$$
or
(40)$$\frac{\partial ln\frac{E_{AE}}{S_{AE}}}{\partial lnA_{av}}=\frac{\partial ln\frac{E}{S}}{\partial lnU_{av}}=\frac{\partial ln\frac{E}{S}}{\partial lnU_{m}}\frac{\partial lnU_{m}}{\partial lnU_{av}}\sim \frac{1}{z}\left(1-\frac{1}{\frac{A_{m}}{C}-1}\right)≅\frac{1}{z}$$
and
(41)$$\frac{\partial ln\frac{E_{AE}}{D}}{\partial lnA_{m}}=\frac{\partial lnE_{AE}}{\partial lnA_{m}}-\frac{\partial lnD}{\partial lnA_{m}}=3-\varphi -\theta ,$$
or
(42)$$\frac{\partial ln\frac{E_{AE}}{D}}{\partial lnA_{av}}=\frac{\partial ln\frac{E_{AE}}{D}}{\partial lnA_{m}}\frac{\partial lnA_{m}}{\partial lnA_{av}}=\frac{1}{z}\left(3-\varphi -\theta \right).$$

The above results can be summarized as follows. The threshold effects lead to the relation between *A_av_* and *A_m_*, as given by Equation (29a). Since the duration time approaches its true value (and $\frac{1}{2ln\frac{A_{m}}{C}}$ goes to zero), only asymptotically for very large values of $\frac{A_{m}}{C}$, $\frac{1}{2ln\frac{A_{m}}{C}}$ can also appear in all relations where the duration time is present. The deviation is less than 5% only if $\frac{A_{m}}{C}≅10^{6}$, and in most of the experiments $\frac{A_{m}}{C}$ can be much less than this limiting value. In addition, since the definition of *A_av_* contains the duration time, there is a transfer correction term, θ, in (29a), (30), and (31), which depends on *A_m_* (and, of course, goes to zero for increasing *A_m_*, i.e., by decreasing the $\frac{\tau_{a}}{T}$ ratio). This term can be calculated from the slope of the experimental $logA_{m}$ versus the logD experimental plot. Relation (24), which expresses the *U_m_*(~*A_m_*)-dependence of the *a* (in Equation (12)) or *B* (in (14) and (23)) proportionality factors, with the exponent *φ*, leads to (25), (32), and (37), which can offer a denouement of the enigmas if φ is close unity. Equation (39) shows that the slope of the $ln\frac{E_{AE}}{S_{AE}}$ versus $lnA_{m}$ indeed should be close to unity (i.e., independent of the parameters $\frac{1}{2ln\frac{A_{m}}{C}}$, *φ*, and *θ*) in accordance with the prediction (10), while the slope of the $ln\frac{E_{AE}}{S_{AE}}$ versus the $lnA_{av}$ plot should be $\frac{1}{z}$ times larger (see Equation (40)). On the contrary, the slope of the $lnE_{AE}$ versus the $lnA_{av}$ plot has a larger slope than that of the $lnE_{AE}$ versus the $lnA_{m}$ plot, providing a smaller deviation than the expected value of 3. It will be shown in the next chapter that when choosing properly the centre of the window of fit on the *A_m_* axis and keeping it fixed for the different scaling plots, the value of *θ*, and, thus, *z*, can be kept constant. Thus, we will obtain that the conclusions drawn from different experimentally determined scaling exponents are in good agreement with each other and are consistent with a *γ* = 2 MF value. 

## 3. Analysis and Discussion of Experimental Data on Scaling Exponents 

Two sets of AE experimental data, obtained on two ferromagnetic shape-memory single crystals, of Ni_45_Co_5_Mn_36.6_In_13.4_ as well as Ni_49_ Fe_18_ Ga_27_ Co_6_ compositions (denoted by alloy A and B, respectively, in the following), during martensitic transformations, will be analysed. The details of the AE measurements on Ni_45_Co_5_Mn_36.6_In_13.4_ are described in [26]. A very similar setup and data acquisition were applied for the AE measurements on the Ni_49_ Fe_18_ Ga_27_ Co_6_ single crystal (the results of which have not been published yet [27]). In both cases, the AE measurements were carried out with Sensophone AED 404 Acoustic Emission Diagnostic Equipment (Geréb and Co., Ltd., Budapest, Hungary) with a piezoelectric sensor (MICRO-100s from Physical Acoustics Corporation, Princeton Junction, NJ, USA). The sampling rate was 16 MHz, and the setup had a band-pass from 30 KHz to 1 MHz. A 30 dB preamplifier and a main amplifier (logarithmic gain) with a 90 dB dynamic range were used. The threshold level was 38 dB, and logarithmic data binning was used. We will just reuse the data obtained in [26,27] for the analysis of the relations predicted in the previous chapter. 

### 3.1. Relations between A_m_ and A_av_, A_m_ and t_m_, and A_m_ and D

Let us first consider the relation between *A_m_* and *t_m_*, since it provides the experimental check of the *A_m_*(*U_m_*)-dependence of the B parameter in (14). Since the *A_m_* and *t_m_* parameters are free of threshold and transfer effects, in this case the fitting can be made from the beginning, up to the upper bound of the fitting window. Figure 1 shows the $logA_{m}$ versus $logt_{m}$ for cooling in alloy A (a) at small, constant, external magnetic field (B = 250 mT) and for heating in alloy B (b) (at B = 0). The slope of this plot is different from unity and is given by Equation (25). It can be seen that, indeed, straight lines are obtained up to certain upper bounds, *A_ub_* = 40 mV and *A_ub_* = 20 mV for alloys A and B, respectively. These values will be used on the *A_m_* axes in the following fits as well. The upper bound, and the scatter of points above it, is most probably caused by the possible overlap of avalanches and by the small numbers of hits at large amplitudes. The rise time, as compared to the duration time, is very short due to the long exponentially decaying tail of the expression (12) and the overlapping of avalanches can result in a reduced effective *A_m_* and increased *t_m_*. From the slopes of the straight lines (2.4 ± 0.1 and 2.2 ± 0.2, for A and B, respectively) φ=0.6∓0.1 and φ=0.6±0.1 are obtained.

Let us now turn to the relation between the maximum and average amplitude, since the power exponent of this scaling relation gives the value of z. Figure 2a,b show these functions for alloy A and B, respectively. It can be seen that, in accordance with the presence of *θ* in Equations (29a) and (31), there is a slight curvature (the slope increases) with increasing $\frac{A_{m}}{C}$ values, and the slopes given in the figure caption belong to the centres of the fitting windows $\frac{A_{m}}{C}=30$ and $\frac{A_{m}}{C}=40$, for the A and B alloys, respectively. These values will be used on the *A_m_* axes in the following fits as well. 

Figure 3 shows the relation between the maximum amplitude and the duration time for alloys A and B, respectively. For alloy B, the curved first part reflects the transfer effects. It is possible to estimate the acoustic-wave-attenuation time (see, e.g., Equation (6) and Figure 8 in [28]) from this part, and $\tau_{a}≅20\;\mu S$ was obtained. Thus, points below about 200 μS can be left out from the fit, since for these $\frac{\tau_{a}}{D}>0.01$. These curves also show an upward curvature, reflecting the *A_m_*-dependence of parameter *θ* (see Equation (30)). It is worth noting that the above three plots in Figure 1, Figure 2 and Figure 3 already provide the values for all the three fitting parameters (*z*, *φ*, and *θ*) used in the previous chapter, for both alloys: z=0.74±0.07, φ=0.6±0.1, and θ = 0.77 ± 0.08 as well as z=0.62±0.07, φ=0.6±0.1, and θ=0.6±0.1, for alloys A and B, respectively. It can be seen that they, taking also into account the error bars of the original exponents, fulfil nicely the predicted relation (31). In the following, from the exponents of other scaling relations, we can collect more data on the above parameters and can obtain their average values too. 

### 3.2. Scaling Relations between E_AE_, S_AE_, $\frac{E_{AE}}{S_{AE}}$, and the Amplitude, A_m_

For these relations, the transfer corrections are negligible, and *θ* is not present in the expressions of the exponents. 

Figure 4 shows the $log\frac{E_{AE}}{S_{AE}}$ versus $logA_{m}$ plots for cooling in alloy A (a) at small, constant, external magnetic field (B = 250 mT) and for heating in alloy B (b) (at B = 0). It can be seen that good straight lines (with slope 0.9 ± 0.1 for A as well as 0.9 ± 0.1 for B, respectively) are obtained. Figure 5 shows for comparison, the $log\frac{E_{AE}}{S_{AE}}$ versus $logA_{av}$ plot for alloy B, which has a slope 1.6 ± 0.1. The slopes of the two straight lines obtained in Figure 4 and Figure 5 for alloy B are in accordance with Equations (39) and (40). The slope of the $log\frac{E_{AE}}{S_{AE}}$ versus $logA_{av}$ plot should be $\frac{1}{z}$ times the one belonging to the $log\frac{E_{AE}}{S_{AE}}$ versus $logA_{m}$ plot: this gives z=0.6∓0.1, which is in a good agreement with z=0.62∓0.07, obtained from Figure 2b. It can be added that the slope of the $log\frac{E_{AE}}{S_{AE}}$ versus $logA_{m}$ plots should be unity seems to be quite a general rule: the slope is indent of *γ*, *φ*, and *θ* (at least for large enough values of *A_m_*) and, thus, can used as a check of the reliability of the AE measurements. 

Figure 6 shows the $logS_{AE}$ versus $logA_{m}$ plots for cooling in alloy A (a) at small, constant, external magnetic field (B = 250 mT) and for heating in alloy B (b) (at B = 0). In the intermediate regions, a straight line can be fitted. The slopes are 1.2∓0.1 and 1.11∓0.05, respectively, and from Equation (32) φ=0.8∓0.1 as well as φ=0.90∓0.08 are obtained. 

Figure 7 shows the $logE_{AE}$ versus $logA_{m}$ plots for cooling in alloy A (a) at small, constant, external magnetic field (B = 250 mT) and for heating in alloy B (b) (at B = 0). The slopes are 2.1∓0.2 and 2.00 ± 0.08, respectively. According to Equation (37), these result in φ=0.9∓0.1 and φ=1.0±0.1, respectively. For comparison, Figure 8 shows the $logE_{AE}$ versus $logA_{av}$ plot for heating in alloy B (b) (at B = 0). The slope of this straight line is 3.7∓0.1, which is in a reasonable accordance with the relation (38): since $\frac{\partial lnE_{AE}}{\partial lnA_{m}}=2.0\mp 0.1$ and z=0.62∓0.07 (see Figure 2) as well as $\frac{\partial lnE_{AE}}{\partial lnA_{av}}=\frac{\partial lnE_{AE}}{\partial lnA_{m}}\frac{1}{z}=3.2\mp 0.3$. 

The results shown in Figure 7 and Figure 8 also illustrate that these results are in very good agreement with the MF value, γ=2. 

### 3.3. Scaling Relation between $\frac{E_{AE}}{D}$ and the Amplitude as Well as between $S_{AE}$ and the Duration Time, D

As it can be seen from Equations (34) and (41), in these cases the transfer correction is not negligible (*θ* is present in the expressions). 

Figure 9 shows the $log\frac{E_{AE}}{D}$ versus $logA_{m}$ plots for alloys A and B, with slopes 1.6 ± 0.1 and 1.50 ± 0.15, respectively. For comparison, Figure 10 shows the $log\frac{E_{AE}}{D}$ versus $logA_{av}$ plot for alloy B, and the slope is 2.6 ± 0.1. Comparing the two slopes obtained in alloy B, we obtain *z* = 0.6 ± 0.1, which is in good agreement with the value obtained from Figure 7 and Figure 8. Furthermore, using the average value of *φ* obtained above (*φ* = 0.8, see also Table 1), θ=0.70 is obtained from Equation (41) for alloy B. 

The most frequently considered scaling relation is $S_{AE}\sim D^{\gamma }$, from which the value of γ was usually calculated and, most frequently, a value less than the MF value, 2, was obtained. Thus, Figure 11 shows the $logS_{AE}$ versus logD plot in alloy B for heating (at B = 0). The slope is sensitive to the window of fit: its value, fitting between 0.2 mS and 3 mS, is γ=2.14±0.17. However, including points belonging to smaller and smaller values of D, the value of *γ* gradually decreases. The γ=2.14 value (which corresponds to the same window of fit used in the previous figures), with the average value of *φ*, where φ=0.8, gives from Equation (34) that *θ* = 0.6, which is in a good agreement with the value obtained above.

We can conclude that all the obtained experimental slopes are in accordance with the relations derived in the previous chapter. The most important parameter is *φ*: this describes the *A_m_*-dependence of the proportionality factor between the two scaling parameters *A_m_* and *t_m_* (see Equations (14) and (23)), and the average values of these for the investigated two alloys are summarized in Table 1. It can be seen that they are the same in both alloys. It is important to emphasize that *φ* = 0.8 can also give account for the observed enigma for the energy-amplitude- and size-amplitude-scaling relations. Furthermore, the obtained results are in very good agreement with the γ=2 MF value: in Equations (32) and (37), the values of 2 and 3 belong to this. 

## 4. Temporal Shape of Avalanches 

Following the proposal of the authors of [25], let us investigate the reduced form of the temporal shape of avalanches at a fixed area, using *A_m_* and *t_m_* as the two scaling parameters, which are not distorted by the transfer properties. Furthermore, as one can expect from Equation (14), these are not independent from each other. Thus, we investigate and compare two cases:(i)assume that B is constant in (14), and both the voltage and time scales will be normalized by *A_m_*;(ii)assume that the scaling parameters are not proportional to each other, but the $\frac{A_{m}}{t_{m}}\sim A_{m}^{\varphi }$ relation holds (see Equations (14), (23) and (24)), i.e., the voltage scale will be reduced by *A_m_* and the time scale by $A_{m}^{1-\varphi }$.

Figure 12 and Figure 13 show the reduced $U^{*}\left(t^{*}\right)$ functions, as an illustration for alloy B, using scaling according to cases (i) and (ii), respectively. It can be seen that in Figure 12, the curves do not fall on a common curve even if only the three curves corresponding to the central part of the fitting window of area, as used in Figure 6, are considered (see the insert). On the other hand, in Figure 13, the curves are scaled much better together (see the insert too), especially in the first, fast-decaying part of the curves. It can be seen that curves belonging to the two first bins (at small values of the fixed area) as well as to the last two bins show some deviation/scatter from the common curve shown in the insert. This can be due to the distortions caused by the transfer and threshold effects (at small values of S) as well as to some overlaps of the large avalanches (for large, fixed values of S). 

## 5. Relation between the Energy and Amplitude for Analysis of Multi-Avalanche Processes 

Finally, it is worth to compare our results to the recent results of [21,22,29]. It was summarized in [29], that in the relation between the energy and amplitude in the form,
(43)$$E=s_{i}A_{i}^{2},$$
the scaling prefactor, $s_{i}$, depends explicitly on the duration of the avalanche, and this can be different for different mechanisms/different avalanche profiles (the index *i* serves to distinguish between groups of signals belonging to different avalanche profiles). It was also emphasized in [28] that, in general, the universality of avalanches contradicts expectations for identifying different underlying physical processes, so it is difficult to specify experimentally measurable parameters that are characteristic for different avalanche profiles. In fact, our formalism would provide a solution for this problem in the following way. If the parameter *φ* depends on the mechanism of the elementary processes generating avalanches, then this parameter will be such a variable. In the present analysis, we acquired the same values for *φ*, but we investigated similar martensitic transformations in two ferromagnetic shape-memory alloys. For the comparison of our result with Equation (43), we should consider our expression (35a), which can be rewritten as
(44)$$E\sim U_{m}^{2}t_{m}I=sU_{m}^{2}$$

It can be seen that the prefactor in (44) is proportional to the $t_{m}I$ product, but instead of using the duration time (which is the most distorted experimental parameter [25]) we use the peak amplitude for the description of the change of this product. Taking, as we have shown above, that the *I* integral is approximately constant, the
(45)$$s\sim U_{m}^{1-\varphi }$$
relation can be obtained. Indeed, if *φ* is different for different avalanche profiles, then Figure 14 shows the $log\frac{E_{AE}}{A_{m}^{2}}$ versus $logA_{m}$ function for alloy A (a) as well as for alloy B (b), under similar conditions as in the previous figures. It can be seen that the slope, (1 − *φ*), is 0.21±0.05 for alloy A, which is in good agreement with the *φ* = 0.77±0.11 value (Table 1). It is worth mentioning that here the fit is more uncertain, although we used a similar fitting window as in the above figures. This can be related to (i) that the value of the slope is small, (ii) that the first part can be distorted due to threshold effects, and (iii) that there is a large scatter at large values of *A_m_* related to overlap of the avalanches and/or to the small number of events per box. Thus, for alloy B, the dashed line is not the line fitted to the points, as it just shows a line with slope 0.17, as expected from *φ* = 0.83 (see Table 1).

## 6. Conclusions

It is shown, using the theoretically predicted temporal-avalanche shape at a fixed area (Equation (12)), that if the voltage scale and the time scale is normalized by the maximum amplitude, *A_m_*, and maximum time (raising time), *t_m_*, then the two scaling parameters are interrelated by $U_{m}\left(\sim A_{m}\right)=Bt_{m}$ (Equation (14)). Here, the parameter B is not constant but can be dependent on *A_m_*. From the analysis of AE measurements on the martensitic transformations in two different single-crystalline shape memory alloys, it was obtained that
(i)from the relation between measured maximum amplitude (*A_m_* ~ *U_m_*) and *t_m_*, the value of *φ* could be determined, and φ=0.73 was obtained (the same values in both alloys);(ii)the *φ* parameter appears in the expression of the power exponents for the relation between the energy and *A_m_* as well as between the area and *A_m_*; these are 3−φ and 2−φ, respectively, which provide a denouement of the enigma;(iii)experimental values of exponents of different scaling relations between the measured AE parameters (energy, area, amplitudes, duration time) are consistent with the above relations;(iv)using *A_m_* and $A_{m}^{1-\varphi }$ parameters for reducing the voltage and time scales, respectively, nice, common temporal-avalanche shapes were obtained for different bins of area.

## Figures and Tables

**Figure 1 materials-15-04556-f001:**
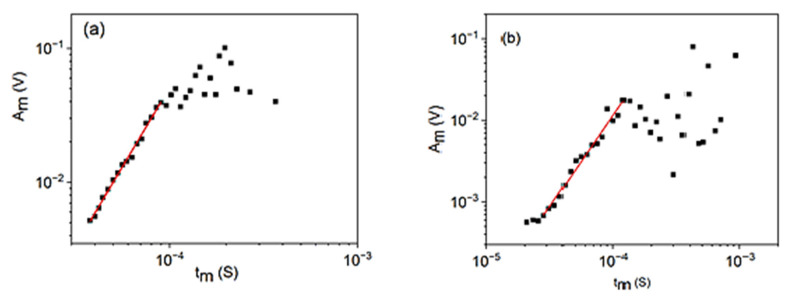
$logA_{m}$ versus $logt_{m}$ plots for cooling in alloy A (**a**) at small, constant, external magnetic field (B = 250 mT) and for heating in alloy B (**b**) (at B = 0).

**Figure 2 materials-15-04556-f002:**
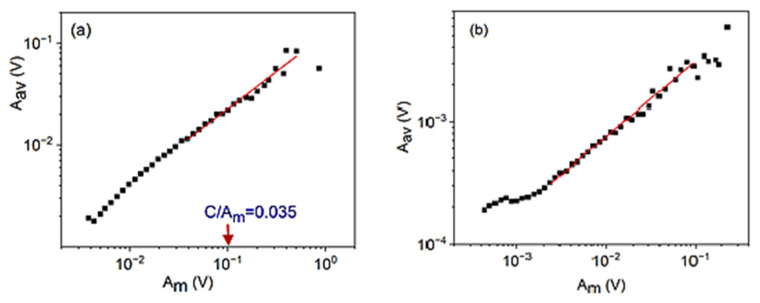
Relation between *A_av_* and *A_m_* for cooling (austenite to martensite) transformation in alloy A (**a**) at small, constant, external magnetic field (B = 250 mT) and for heating in alloy B (**b**) (at B = 0). The slopes and the values of the centre of the fits are z=0.74∓0.07 and $\frac{A_{m}}{C}=30$ (C = 3.5 mV), as well as z=0.62∓0.07 and $\frac{A_{m}}{C}=40$ (C = 0.4 mV), for A and B, respectively.

**Figure 3 materials-15-04556-f003:**
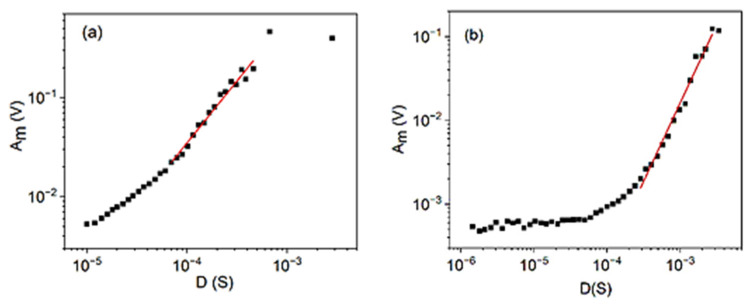
$logA_{m}$ versus logD plots for cooling in alloy A (**a**) at small, constant, external magnetic field (B = 250 mT) and for heating in alloy B (**b**) (at B = 0). The slopes (1.3 ± 0.1 and 1.9 ± 0.1, respectively), using again the same fitting windows with the same mid values of *A_m_* as in Figure 2, provide $\theta =\frac{1}{1.3}=0.77\mp 0.08$ and θ=0.6±0.1 for alloys A and B, respectively.

**Figure 4 materials-15-04556-f004:**
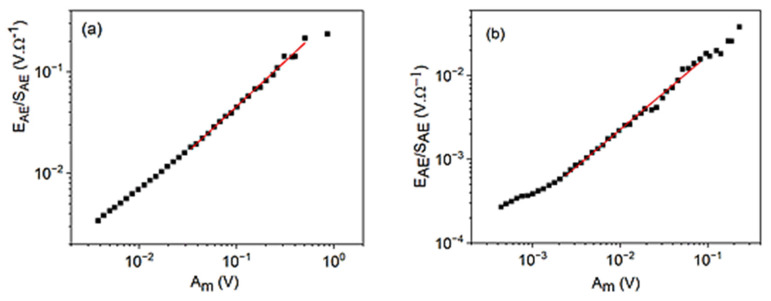
$log\frac{E_{AE}}{S_{AE}}$ versus $logA_{m}$ plots for cooling in alloy A (**a**) at small, constant, external magnetic field (B = 250 mT) and for heating in alloy B (**b**) (at B = 0).

**Figure 5 materials-15-04556-f005:**
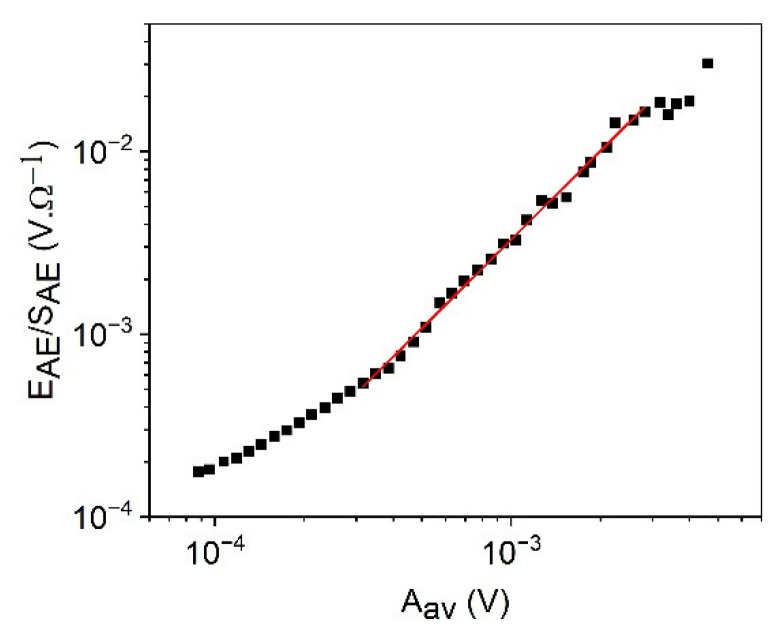
$log\frac{E_{AE}}{S_{AE}}$ versus $logA_{av}$ plot for heating in alloy B (at B = 0).

**Figure 6 materials-15-04556-f006:**
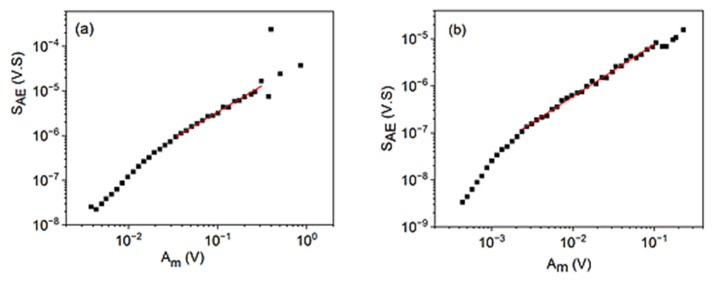
$logS_{AE}$ versus $logA_{m}$ plots for cooling in alloy A (**a**) at small, constant, external magnetic field (B = 250 mT) and for heating in alloy B (**b**) (at B = 0).

**Figure 7 materials-15-04556-f007:**
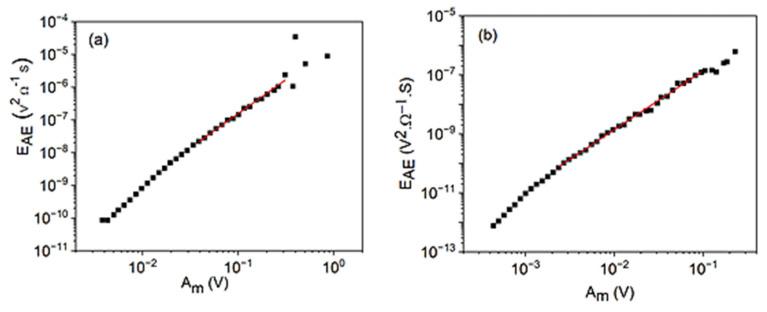
$logE_{AE}$ versus $logA_{m}$ plots for cooling in alloy A (**a**) at small, constant, external magnetic field (B = 250 mT) and for heating in alloy B (**b**) (at B = 0).

**Figure 8 materials-15-04556-f008:**
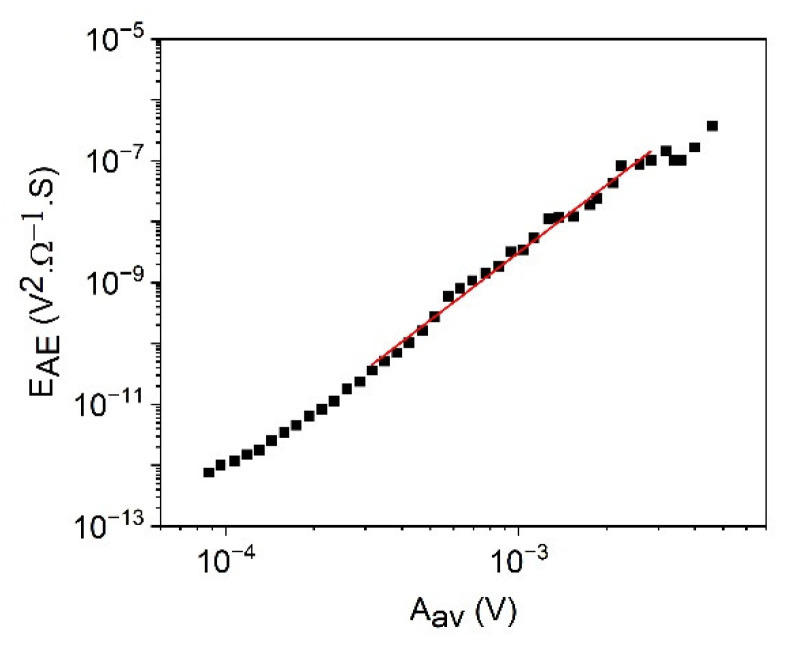
$logE_{AE}$ versus $logA_{av}$ plot for heating in alloy B (at B = 0).

**Figure 9 materials-15-04556-f009:**
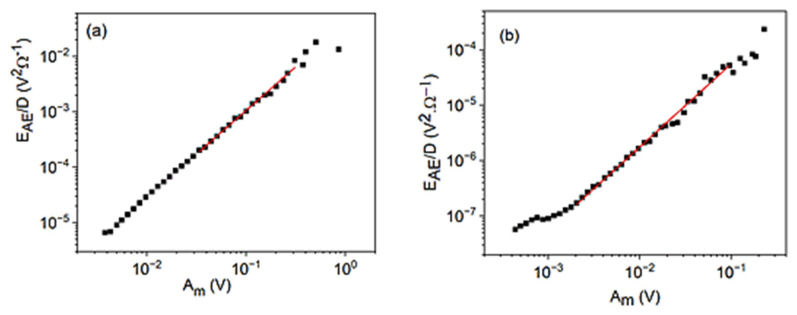
$log\frac{E_{AE}}{D}$ versus $logA_{m}$ plots for cooling in alloy A (**a**) at small, constant, external magnetic field (B = 250 mT) and for heating in alloy B (**b**) (at B = 0). The slopes are 1.6 ± 0.1 and 1.65 ± 0.05, respectively.

**Figure 10 materials-15-04556-f010:**
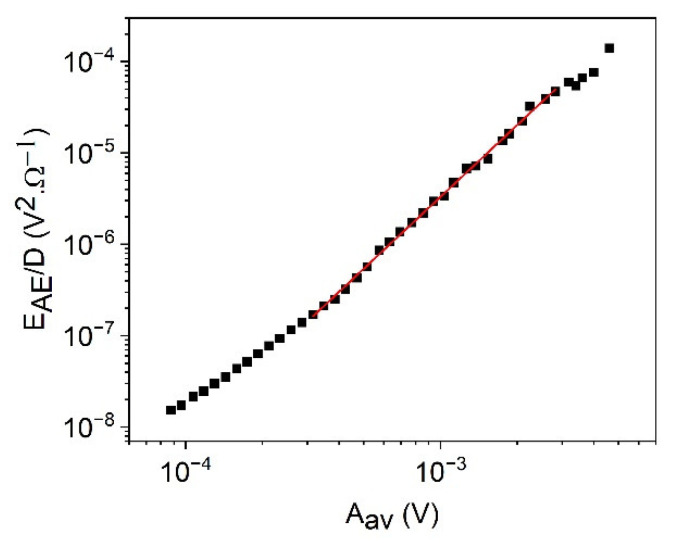
$log\frac{E_{AE}}{D}$ versus $logA_{av}$ plot for heating in alloy B (at B = 0). The slope is 2.30 ± 0.05.

**Figure 11 materials-15-04556-f011:**
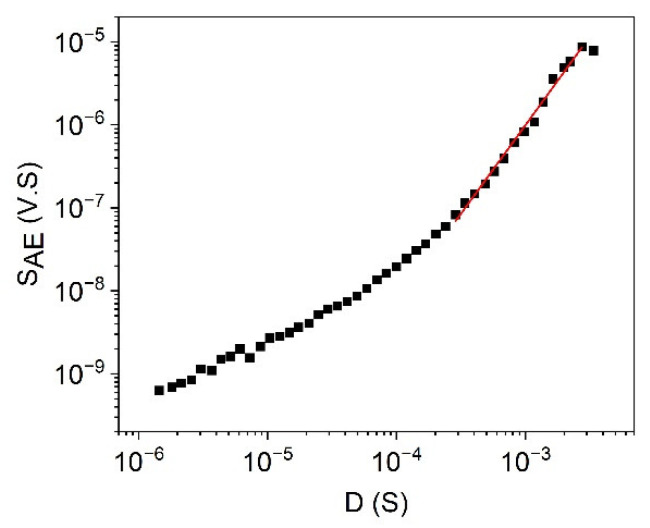
$logS_{AE}$ versus logD plot for heating in alloy B (at B = 0). The slope is sensitive to the window of fit: e.g., fitting between 0.2 mS and 3 mS, the slope is 2.14±0.17.

**Figure 12 materials-15-04556-f012:**
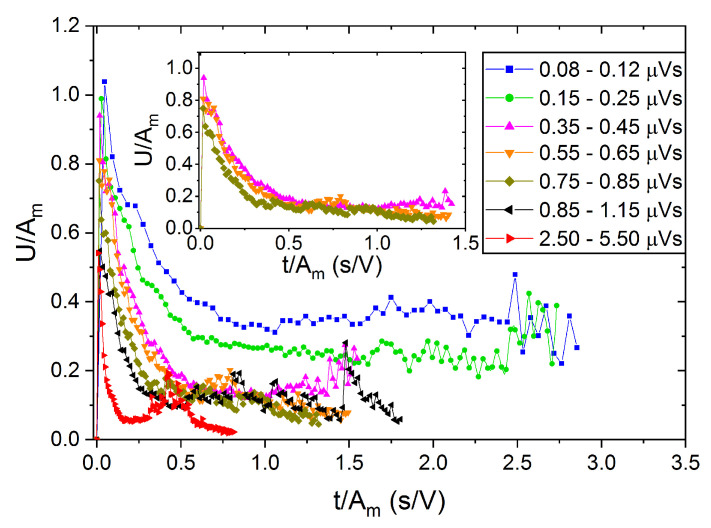
Normalized $U^{*}\left(t^{*}\right)$ functions obtained by scaling both the voltage and time scales by the peak amplitude, *A_m_*, for heating in alloy B (at B = 0) at different bins of avalanche area, S.

**Figure 13 materials-15-04556-f013:**
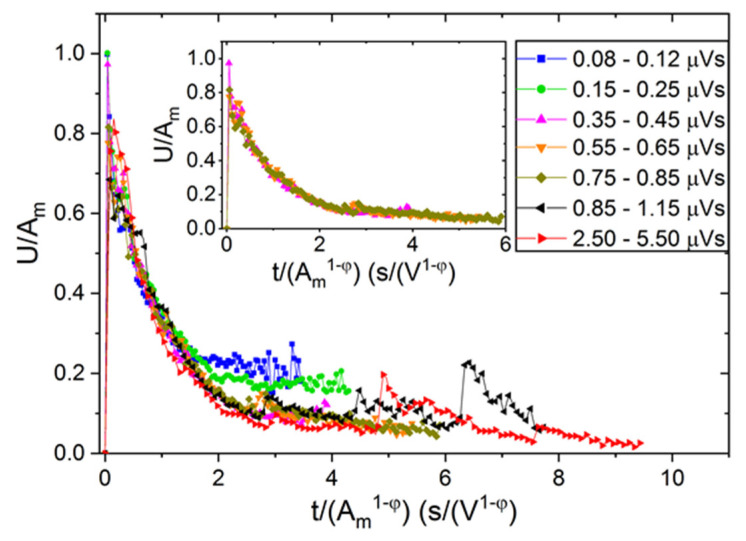
Normalized $U^{*}\left(t^{*}\right)$ functions obtained by scaling the voltage with *A_m_* and time scales by $A_{m}^{1-\varphi }$ (*φ* = 0.74: see Table 1) for heating in alloy B (at B = 0) at different bins of avalanche area, S.

**Figure 14 materials-15-04556-f014:**
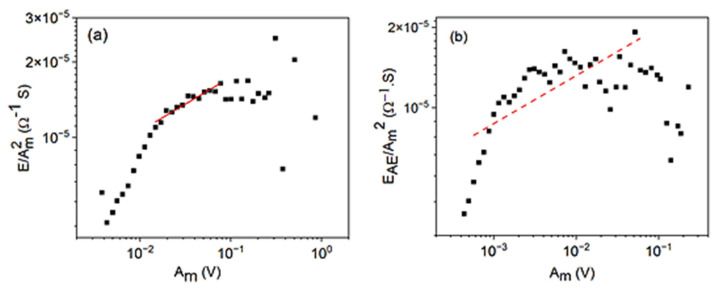
$log\frac{E_{AE}}{A_{m}^{2}}$ versus $logA_{m}$ functions for cooling in alloy A (**a**) at small, constant, external magnetic field (B = 250 mT) and for heating in alloy B (**b**) (at B = 0 mT).

**Table 1 materials-15-04556-t001:** Values of φ calculated from different relations for alloy A and B, respectively. It can be seen that they are very similar, and the average value for the two alloys is *φ_av_* ≅ 0.8 ± 0.1.

Equation	Value of *φ*	
	Alloy A	Alloy B
(25)	0.6 ± 0.1	0.6 ± 0.1
(32)	0.8 ± 0.1	0.90 ± 0.08
(37)	0.9 ± 0.1	1.0 ± 0.1
Average	0.77 ± 0.11	0.83 ± 0.13

## Data Availability

The data that support the findings of this study are available from the corresponding author upon reasonable request.

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
