# Peer review of "Denouement of the Energy-Amplitude and Size-Amplitude Enigma for Acoustic-Emission Investigations of Materials"

_materials, 2022, doi:10.3390/ma15134556_

Round 1
Reviewer 1 Report
Sarah M. Kamel et al. have shown that normalizing U by A and t by rise time the exponent of the logtm versus logA plot is given by 1-φ, with material independent φ. Furthermore, they get that ?~?3−? and ?~?2−?, providing denouement of the enigmas. Some findings have been emphasizd and some statements have been revealed accordingly. In my best opinion, the paper is informative and well organized. It can be accepted after Minor Revision. The reviewer's comments are below:
1) I recommend to enrich the introduction with some relevant works in the literature in order to guide the material readers.
2) In page 3, line 98 the equation should be numbered. So, all equation should be renumbered throughout the manuscript. The same remark in page 6 line 220. Page 7 line 278 & 280. Table 1 can be affected.
3) Proofreading by a native speaker can improve the narative and some language issues.
Good Luck
Reviewer 2 Report
1. It is suggested that the Abstract should be condensed, and more emphasis should be given on the added value of this work on it and Introduction by pointing out what is missing from the already existent studies.
2. There are many derivations or formulas in this article. It is suggested that important parameters should be stated or explained graphically. In doing this, it will be easier for the reader to understand the meaning that the authors want to express. Furthermore, are there any definitional conflicts between the derivations or formulas cited in different references?
3. This article used two sets of AE experimental data, obtained on two ferromagnetic shape memory single crystals, of Ni45Co5Mn36.6In13.4 as well as Ni49 Fe18 Ga27 Co6 compositions for analytical work. Because of the importance of the data, some further clarification about this experiment is needed, not just the cited references in the article.
4. How to prove that the results of Figures 1 to 11 are correct without deduction and calculation errors? What is the reason for the non-straight lines on the right side of Figure 1?
5. Section 4, Temporal shape of avalanches: are there any limitations to the two assumptions proposed, B and scaling parameters?
6. Conclusion should be shorter and more precise.
Reviewer 3 Report
Authors investigate the acoustic emission of materials with respect to the dependence of the detected voltage signal and а fixed area on the peak amplitude of crackling noise.
Paper is well written, based on an idea to correct the known relations to make them fit better with practical uses. For the demonstration, authors apply their expressions on AE measurements of martensitic transformations in single crystalline shape memory alloys. Claims are supported and discussed clearly.
The topic is of interest to the readership and fits into the scope of the MDPI journal Materials. Some place for improvement is in the domain of better readability and layout.
For instance, annotation in Fig 12 and 13 (and their sub-figures) is easier to understand if the size is comparable to the font size of the main text.
Some of typos are:
line 32 - "Um" shouldn't be in the list but separately explained, apart from the PDF definition
line 67 - "no" should be "not"
line 549 - "From" instead of "From"...
Overall recommendation is acceptance.
Thank you for contributing to this journal and science with this work.
With kind regards…
Reviewer 4 Report
Review report on the topic ‘Denouement of the energy-amplitude and size-amplitude enigma for acoustic emission investigations of materials’. Comments are listed below:
1. Strengthen the abstract section. Add the key conclusion of the works in the last two lines of the abstract section.
2. Discuss the motive behind the work. The clear application of the work should be discussed in the introduction section. From the introduction section application of the work is not clear.
3. There are numerous spelling and grammatical errors. Please revise the manuscript thoroughly. Sentences are also not complete.
4. The novelty of the work should also be discussed in a separate paragraph.
5. Try to make a bridge between current and previously published work and specify the gap area and objective of the work. Add the specific gap observed from the literature at the end of the introduction section.
6. The manuscript is written very poorly. A lot of English and grammatical errors in the manuscript. Please improve the quality of the writing.
7. Technical discussion is very poor and a number of already derived equations are presented without any reference.
Round 2
Reviewer 2 Report
See the attached file.

Reviewer 4 Report
Accepted.